# Cryopreservation of Fetal Porcine Kidneys for Xenogeneic Regenerative Medicine

**DOI:** 10.3390/jcm12062293

**Published:** 2023-03-15

**Authors:** Kenji Matsui, Yoshitaka Kinoshita, Yuka Inage, Naoto Matsumoto, Keita Morimoto, Yatsumu Saito, Tsuyoshi Takamura, Hitomi Matsunari, Shuichiro Yamanaka, Hiroshi Nagashima, Eiji Kobayashi, Takashi Yokoo

**Affiliations:** 1Division of Nephrology and Hypertension, Department of Internal Medicine, The Jikei University School of Medicine, Tokyo 105-8461, Japan; 2Department of Urology, Graduate School of Medicine, The University of Tokyo, Tokyo 113-8654, Japan; 3Department of Pediatrics, The Jikei University School of Medicine, Tokyo 105-8461, Japan; 4Meiji University International Institute for Bio-Resource Research, Kanagawa 214-8571, Japan; 5Department of Kidney Regenerative Medicine, The Jikei University School of Medicine, Tokyo 105-8461, Japan

**Keywords:** fetal pig, kidney xenotransplantation, vitrification, organ preservation

## Abstract

Kidney xenotransplantation has been attracting attention as a treatment option for end-stage renal disease. Fetal porcine kidneys are particularly promising grafts because they can reduce rejection through vascularization from host vessels. We are proposing xenogeneic regenerative medicine using fetal porcine kidneys injected with human nephron progenitor cells. For clinical application, it is desirable to establish reliable methods for the preservation and quality assessment of grafts. We evaluated the differentiation potency of vitrified porcine fetal kidneys compared with nonfrozen kidneys, using an in vivo differentiation model. Fetal porcine kidneys connected to the bladder were frozen via vitrification and stored in liquid nitrogen. Several days later, they were thawed and transplanted under the retroperitoneum of immunocompromised mice. After 14 days, the frozen kidneys grew and differentiated into mature nephrons, and the findings were comparable to those of nonfrozen kidneys. In conclusion, we demonstrated that the differentiation potency of vitrified fetal porcine kidneys could be evaluated using this model, thereby providing a practical protocol to assess the quality of individual lots.

## 1. Introduction

Kidney transplantation is the only treatment option for end-stage renal disease, but few people can benefit owing to a severe organ shortage [1]. Recent advances in genetic modification technology for pigs have attracted attention to xenotransplantation [2,3], but the effect of strong immunosuppression on patients is a concern. Our group is independently developing “xenogeneic regenerative medicine” that involves the use of porcine metanephroi (MNs; fetal kidneys) injected with human nephron progenitor cells (NPCs). This approach has some advantages. First, host blood vessels invade the MNs, avoiding the vascular-type hyperacute rejection that is observed in adult porcine kidney transplantation [4]. Second, this method uses the developmental niche of MNs, which may allow human NPCs to form chimeric nephrons connected to the porcine nephrons [5,6,7]. Accordingly, the vessels and partial nephron structures will be humanized, making immunosuppression easier [4,8].

For clinical application, properly preserved, ready-to-use grafts may be useful because the timing of transplantation must be adjusted according to the patient’s condition. We have previously demonstrated that cryopreserved fetal pig [9] and mouse [10] MNs differentiated well. However, an effective method to evaluate vitrified MNs has not been investigated thus far. In this study, we demonstrated that the differentiation potency of vitrified porcine MNs could be evaluated using an in vivo differentiation model.

## 2. Materials and Methods

### 2.1. Research Animals

All animals were treated in accordance with the Guidelines for Proper Conduct of Animal Experiments. The animal studies were approved by the animal ethics committee of The Jikei University School of Medicine (approval number: 2021-069). Pregnant microminiature pigs (MMPs; Fuji Micra, Inc., Shizuoka, Japan) were used to extract MNs connected to the bladder (MNBs) from fetal pigs. Male NOD/Shi-scid, IL-2RgKO Jic mice (NOG mice; CLEA Japan, Inc., Tokyo, Japan) were used as recipients.

### 2.2. Collection of Fetal Porcine Kidneys

On embryonic day 30 (E30), pregnant MMPs were administered general anesthesia, and a laparotomy was performed through an abdominal midline incision. The fetuses were extracted from the exposed uterus, promptly decapitated, and transported on ice in 5-mL tubes (30119401; Eppendorf, Hamburg, Germany) containing minimum essential medium α (MEM α; 12561-056; Invitrogen, Waltham, MA, USA), which had been pre-equilibrated with a 5% CO_2_ atmosphere. This was based on a report that fetal mouse kidneys stayed viable after keeping decapitated fetuses at 0 °C or 4 °C for 72 h [11]. The fetuses were transported to another laboratory to extract MNBs under a stereomicroscope (M205FA; Leica Microsystems, Wetzlar, Germany). Maternal pigs were kept normally after closing the uterus and the abdomen.

### 2.3. Cryopreservation of Fetal Porcine Kidneys

MNBs were cryopreserved via vitrification as previously reported [9,10,12]. First, the MNBs were equilibrated in base medium (MEM α supplemented with 20% fetal bovine serum (FBS; SH30070.03, HyClone Laboratories, Inc., Logan, UT, USA) and 1% antibiotic–antimycotic solution [15,240,062; Thermo Fisher Scientific, Waltham, MA, USA]) with 7.5% ethylene glycol (EG; 055-00996; Wako, Osaka, Japan) and 7.5% dimethyl sulfoxide (DMSO; 317275-100ML; Millipore, Burlington, MA, USA) on ice for 15 min and soaked in base medium with 15% EG and 15% DMSO on ice for an additional 15 min. Next, the MNBs were placed on Cryotops (81111; Kitazato Corporation, Tokyo, Japan) and directly plunged into liquid nitrogen. They were stored in liquid nitrogen tanks until transplantation. The time spent on each step was recorded.

### 2.4. Thawing of Fetal Porcine Kidneys

MNBs cryopreserved for 3 days or more were thawed just before transplantation. The Cryotops with MNBs were quickly transferred from liquid nitrogen to base medium with 1 M sucrose at 42 °C for 1 min, transferred to base medium with 0.5 M sucrose at room temperature for 3 min, and finally washed twice in base medium at room temperature for 5 min each time. The fetal kidneys were separated from the bladder and trimmed under a stereomicroscope.

### 2.5. Transplantation of Fetal Kidneys under the Retroperitoneum of NOG Mice

NOG mice were anesthetized using isoflurane inhalation (2817774; Pfizer, New York, NY, USA), and a laparotomy was performed through an abdominal midline incision. A pocket was created in the retroperitoneum in the area bounded by the aorta, left ureter, and left renal artery, using micro-tweezers (11253-25; Dumont, Montignez, Switzerland) under a surgical microscope (S9D; Leica Microsystems). One fetal kidney per recipient was transplanted into the pocket, which was then closed with 10-0 nylon thread (Muranaka Medical Instruments Co. Ltd., Osaka, Japan). The operation was completed by closing the abdomen. As a control, nonfrozen fetal kidneys extracted from E30 fetal pigs were transplanted into NOG mice on the day of extraction. Fourteen days after transplantation, the NOG mice were euthanized via cervical dislocation, and the grafts were harvested.

### 2.6. Hematoxylin-and-Eosin Staining and Immunostaining of Paraffin Sections

The harvested kidneys were fixed in 4% paraformaldehyde (161-20141, Wako) at 4 °C overnight and embedded in paraffin. Long-axis sections with a thickness of 5 µm were prepared from the whole kidneys. Three sections from each of the first, median, and third quartiles of thickness were selected for hematoxylin-and-eosin (HE) staining using standard procedures. For immunostaining, the slides were deparaffinized, washed thrice in phosphate-buffered saline (PBS), and incubated with citrate buffer at 121 °C for 10 min for antigen retrieval. After again washing thrice in PBS, the slides were blocked at room temperature for 10 min using Blocking One Histo (06349-64; Nacalai Tesque, Kyoto, Japan). Subsequently, the slides were incubated overnight at 4 °C with primary antibodies, washed thrice in PBS, and incubated with secondary antibodies at room temperature for 1 h. Finally, the slides were washed thrice in PBS and mounted in Prolong Gold Antifade Mountant with DAPI (P36931; Invitrogen). All samples were evaluated under a fluorescence microscope (LSM880; Carl Zeiss, Oberkochen, Germany). Regarding the primary antibodies, we used anti-dachshund family transcription factor 1 (DACH1; 10914-1-AP; Proteintech, Rosemont, IL, USA) for podocytes, anti-lotus tetragonolobus lectin (LTL; B-1325; Vector Laboratories, Burlingame, CA, USA) for the proximal tubules, anti-E-cadherin (ECAD; 3195S; Cell Signaling Technology, Danvers, MA, USA) for the distal tubules, anti-cytokeratin 8 (CK8; TROMA-I-C; Developmental Studies Hybridoma Bank, Iowa City, IA, USA) for the collecting ducts, and anti-sine oculis homeobox homolog 2 (Six2; 11562-1-AP; Proteintech) for NPCs.

### 2.7. Statistical Analysis

All data are presented as means ± standard errors of the mean (SEM). Data were analyzed using the two-tailed Mann–Whitney U test; A *p* value of <0.05 was considered statistically significant. The data were analyzed using GraphPad Prism software, version 8.0 (GraphPad Software, San Diego, CA, USA).

## 3. Results

### 3.1. Gross Characteristics of the Fetal Porcine Kidneys before and after Transplantation

A schematic of the freezing process is provided in Figure 1A. E30 porcine fetuses were decapitated 7 ± 1 min after extraction from the uterus, and the time of transportation was 283 ± 11 min. The MNBs were extracted (Figure 1B), vitrified, and frozen in 57 ± 3 min. Either frozen or nonfrozen fetal kidneys were transplanted into the retroperitoneum of NOG mice (Figure 1C). Fourteen days after transplantation, the kidneys had increased in size and were reddish, suggesting invasion of host blood vessels (Figure 1D). The average size of the frozen fetal kidneys (n = 8; originally, 1.37 × 0.90 mm) increased by 2.1 and 2.2 times in the long and short diameters, respectively; these findings were not much different from those of the nonfrozen fetal kidneys (n = 2) (Figure 1E,F).

### 3.2. Quantitative Evaluation of Kidneys with Hematoxylin-and-Eosin Staining

The harvested kidneys were sectioned along the long axis, and three sections from each of the first, median, and third quartiles of thickness were subjected to HE staining (Figure 2A,B). The number of mature glomeruli in each section was counted, and the average of the three slides was calculated. The frozen kidneys had more than four glomeruli per slide (n = 4), which was comparable to the finding of the nonfrozen kidneys (n = 2) (Figure 2C). In both the frozen and nonfrozen kidneys, the medulla was a sparse, connective tissue-dominated component, compared with native neonatal kidneys (Figure 2B,D).

### 3.3. Detailed Structural Evaluation by Immunostaining

Immunostaining was conducted to assess the degree of development of the frozen kidneys in detail compared with the nonfrozen kidneys. Staining with mature nephron markers confirmed the presence of glomeruli, proximal and distal tubules, and collecting ducts in both kidneys (Figure 2E–H). In addition, Six2-positive NPCs that surround CK8-positive ureteric buds remained in a part of the cortex, suggesting continued renal development (Figure 2G,H).

## 4. Discussion

In recent years, xenotransplantation using porcine organs has attracted attention, and transplantation of porcine kidneys into humans with rejection control through multiple genetic manipulations has been reported [2,3]. On the other hand, studies regarding the transplantation of fetal porcine pancreatic tissue into humans [13] and fetal porcine kidneys into cynomolgus monkeys [4] have shown that fetal porcine organs can reduce rejection because they are vascularized by host vessels.

For the clinical application of fetal organs, properly preserved, ready-to-use grafts may be useful. In the vitrification method, intracellular water is replaced with highly concentrated cryoprotectants, such as DMSO and EG, before quick freezing to −196 °C in order to render the cytoplasm amorphous, preventing cell damage due to ice crystal formation [14,15]. Details have been reported using mouse embryos [16], and this approach is currently used for human oocytes and ovarian tissue sliced 1-mm thick [15,17]. We were the first to apply this method to fetal kidneys and have transplanted vitrified fetal porcine kidneys into pigs and monkeys [4,9]. However, the effect of vitrification on the post-transplantation viability of MNs, especially on proper kidney differentiation and development, required further elucidation.

In the present study, frozen and nonfrozen fetal porcine kidneys were transplanted into immunocompromised mice. After 14 days, all frozen kidneys in four independent procedures formed mature glomeruli and tubules, and the findings were comparable to those of the nonfrozen kidneys. Therefore, this method appears to be reliable for the preservation of fetal porcine kidneys. We have also found that this in vivo differentiation model can be used to verify the quality of individual lots.

In this study, two additional findings were obtained. First, the transplanted kidneys increased in size and NPCs remained for at least 14 days, indicating that renal development was continuing. This may support the usefulness of xenogeneic regenerative medicine, in which fetal organs are allowed to grow and gain function in the host body. In contrast, the previous report showed that transplanted E12.5 fetal mouse kidneys lost the expression of markers for NPCs after 12 days [18], suggesting that differences in developmental rates among species might persist after transplantation. Second, the medulla of the transplanted kidneys seemed relatively sparse, and the collecting ducts were not arranged radially as in native neonatal kidneys. This is consistent with the previous report that the collecting ducts were relatively immature in transplanted fetal mouse kidneys compared with more proximal segments [18].

As this study was conducted over a relatively short period of time, further verification is needed to determine the extent to which the transplanted fetal kidneys mature and function. In addition, for future clinical application, it will be necessary to develop a process to collect and preserve fetal organs under germ-free and designated pathogen-free conditions.

## 5. Conclusions

We demonstrated that the differentiation potency of vitrified fetal porcine kidneys could be evaluated using an in vivo differentiation model.

## Figures and Tables

**Figure 1 jcm-12-02293-f001:**
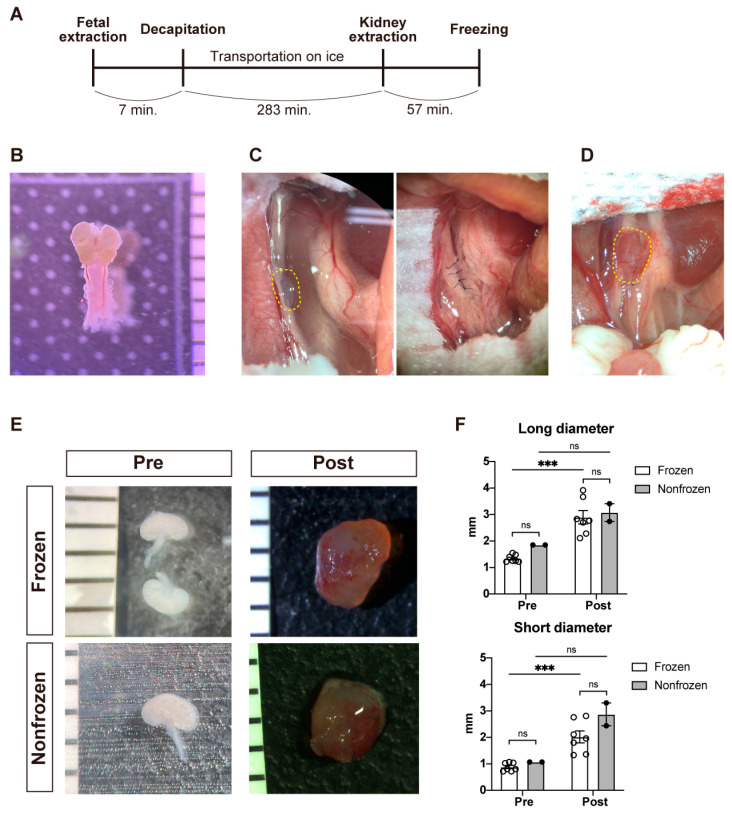
Gross characteristics of the transplanted fetal kidneys. (**A**) A schematic showing the time course of fetal pig collection, decapitation, fetal transportation on ice, kidney collection, and vitrification and freezing. (**B**) Fetal kidneys and bladder from a fetus of a microminiature pig on embryonic day 30. (**C**) Macroscopic view of the transplantation of a fetal kidney into an adult NOD/Shi-scid, IL-2RgKO Jic (NOG) mouse. The retroperitoneum on the left side of the aorta was incised to create a pocket, one kidney was transplanted (**left**), and the incision was closed with 10-0 nylon (**right**). (**D**) Macroscopic view of the recipient NOG mouse 14 days after transplantation. The kidney is enlarged and reddish. (**E**) Macroscopic view of the frozen (**left**) and nonfrozen (**right**) fetal kidneys before and after transplantation. (**F**) Long and short diameters of the frozen (n = 8) and nonfrozen (n = 2) kidneys before and after transplantation. The circles represent individual specimens. The error bars in the bar plots represent SEM. *** *p* < 0.001; ns, not significant.

**Figure 2 jcm-12-02293-f002:**
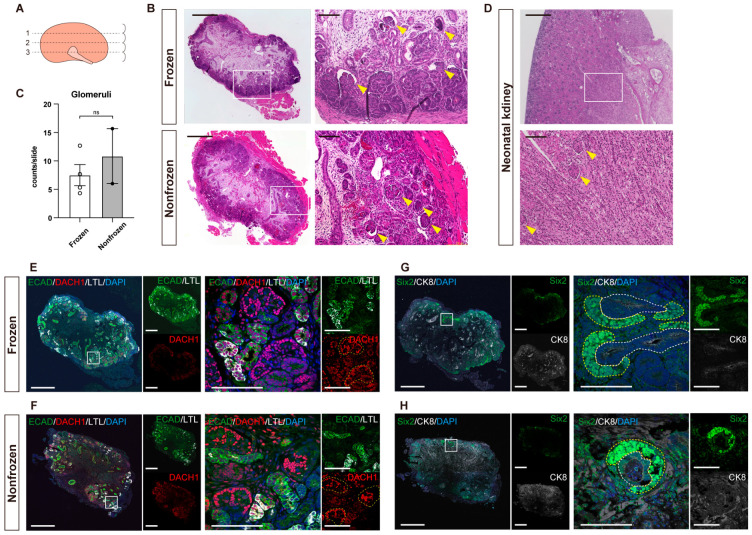
Assessment of the maturity of the transplanted frozen kidneys. (**A**) A schematic of section selection of the harvested kidneys. (**B**,**D**) Hematoxylin-and-eosin staining of the transplanted frozen and nonfrozen fetal kidneys (**B**) and native neonatal kidneys on E111 (**D**). Glomeruli are indicated by yellow arrowheads. (**C**) Mean number of glomeruli per section of frozen (n = 4) and nonfrozen (n = 2) kidneys. The circles represent individual specimens; ns, not significant The error bars in the bar plots represent SEM; ns, not significant. (**E**–**H**) Immunostaining of the transplanted frozen kidneys (**E**,**G**) and nonfrozen kidneys (**F**,**H**). (**E**,**F**) Staining for DACH1 (podocytes), LTL (proximal tubules), and ECAD (distal tubules). Glomeruli are indicated by yellow dotted lines. (**G**,**H**) Staining for Six2 (nephron progenitor cells; NPCs) and CK8 (ureteric buds and collecting ducts). Developing renal structures that consist of ureteric bud (white dotted lines) and surrounding NPCs (yellow dotted lines) exit in a part of the cortex. Scale bars: 500 μm in (**B left**, **D upper**, and **E**–**G**
**left**); 100 μm in (**B right**, **D lower**, and **E**–**G**
**right**). ECAD, E-cadherin; DACH1, dachshund family transcription factor 1; LTL, lotus tetragonolobus lectin; Six2, sine oculis homeobox homolog 2; CK8, cytokeratin 8; DAPI, 4′,6-diamidino-2-phenylindole.

## Data Availability

All relevant data supporting the findings of this study are either included within the article or are available upon request from the corresponding author.

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
