# Peer review of "Cryopreservation of Fetal Porcine Kidneys for Xenogeneic Regenerative Medicine"

_jcm, 2023, doi:10.3390/jcm12062293_

Round 1

Reviewer 1 Report

The manuscript by Matsui et al. describes the transplantation of porcine foetal kidneys (day 30, metanephric kidneys) into immune deficient mice, and comparing the development of cryopreserved versus freshly isolated foetal kidneys over a 14-day period. Characterisation included size, histology, and staining for Six-2  (detection of multipotent self-renewing nephron progenitors).     

Their results show successful transplantation, that both kidneys developed at a similar rate, the presence of glomeruli, proximal and distal tubules, and collecting ducts, which were less regular than seen in adult kidneys.

Overall there was no significant difference between the cryopreserved and non-cryopreserved foetal kidneys. A comparison with normal kidney development within the porcine foetus is not provided, which might be important to show.

Regarding novelty: There has been a number of transplant experiments of porcine foetal kidneys into mice and detailed characterisation. The authors themselves have transplanted cryopreserved foetal kidneys into immune competent monkeys over a much longer period of time and showed their viability and development. This has been a more informative experiment considering the overall goal, transplant into human patients.

This said, the manuscript provides direct evidence that at least short-term cryopreservation has no detrimental effects. In addition the authors suggest that compared to some previous experiments, they could show the presence of nephron progenitors and that it might be method to assess the quality of a batch of frozen foetal kidneys prior to transplant into humans. But it can not determine functionality.

The methods used are technically very challenging. and figures confirm the described results.

The paper is well written, and requires only minor clarifications.

1.      Line 95: “…nonfrozen fetal kidneys extracted from E30 fetal pigs on the same day were used“ does this mean they were isolated on the same day as the cryopreserved kidneys and then cultured for 3 days. Or: isolated and transplanted on the same day? Please clarify. If they have been cultured, could that have a negative effect?

2.      Line 179: “For the clinical application of fetal organs, grafts must be preserved...” It might be convenient to have readily available grafts, but it is not a must. Please rephrase.

In summary: Well executed experiments, providing limited novel information. However, providing direct evidence, that cryopreservation has no detrimental effect, is important for the field.

Author Response

Reviewer 1

Point 1:

A comparison with normal kidney development within the porcine foetus is not provided, which might be important to show.

Response 1: Thank you for your comment. We have added HE staining of a kidney from a native neonatal pig (a kidney grown inside a pig fetus) to Figure 2D.

Point 2:

Line 95: “…nonfrozen fetal kidneys extracted from E30 fetal pigs on the same day were used“ does this mean they were isolated on the same day as the cryopreserved kidneys and then cultured for 3 days. Or: isolated and transplanted on the same day? Please clarify. If they have been cultured, could that have a negative effect?

Response 2: The nonfrozen kidneys were isolated and transplanted on the same day. The sentence has been revised for clarity.

Point 3:

Line 179: “For the clinical application of fetal organs, grafts must be preserved...” It might be convenient to have readily available grafts, but it is not a must. Please rephrase.

Response 3: According to this comment, we have rephrased the sentences in Discussion as well as in Introduction.

Reviewer 2 Report

In the present study xenogeneic regenerative medicine was presented using fetal porcine kidney grafts injected with human nephron progenitor cells. The differentiation potency of vitrified porcine fetal kidneys was compared to nonfrozen kidneys, using an in vivo differentiation model.

The frozen porcine kidneys were transplanted under the retroperitoneum of immunocompromised mice. The nephrons of frozen kidneys grew and differentiated into mature forms the same way as in nonfrozen kidneys.

The authors demonstrated that the differentiation potency of vitrified fetal porcine kidneys could be evaluated using an in vivo differentiation model.

All sections of the manuscript are well written. However, as experienced histologist I would like to see more labelling in Fig.3: which structures are present in differentiating kidneys. In Fig.3D the histological structure presented as belonging to kidney appears like an oocyte, therefore detailed labelling could improve understanding of the microphotographs shown in the Results section.

Author Response

Reviewer 2

Point 1:

All sections of the manuscript are well written. However, as experienced histologist I would like to see more labelling in Fig.3: which structures are present in differentiating kidneys. In Fig.3D the histological structure presented as belonging to kidney appears like an oocyte, therefore detailed labelling could improve understanding of the microphotographs shown in the Results section.

Response 1: Thank you for your comment. In Figures 2G and 2H that were 3B and 3D before revision, the developing renal structures that consist of ureteric bud (CK8+) and surrounding nephron progenitor cells (Six2+) were emphasized by dotted lines. In addition, glomeruli in Figures 2E and 2F were also highlighted. These highlights were noted in the legend.
